# High Sensitivity Strain Sensors Using Four-Core Fibers through a Corner-Core Excitation

**DOI:** 10.3390/mi13030431

**Published:** 2022-03-11

**Authors:** Lina Suo, Ya-Pei Peng, Cheng-Kai Yao, Shijie Ren, Xinhe Lu, Nan-Kuang Chen

**Affiliations:** 1School of Physics Sciences and Information Technology, Liaocheng University, Liaocheng 252000, China; 1920110506@stu.lcu.edu.cn (L.S.); renshj@tom.com (S.R.); 2College of Engineering Physics, Shenzhen Technology University, Shenzhen 518000, China; pengyapei@sztu.edu.cn; 3NK Photonics Ltd., Jinan 250119, China; xinhelu@nkphotoincs.com; 4Department of Electro-Optical Engineering, National Taipei University of Technology, Taipei 10608, Taiwan; pyps41520@gmail.com

**Keywords:** multicore fiber, four core fiber, supermodes, strain sensors, asymmetric modes

## Abstract

A weakly-coupled multicore fiber can generate supermodes when the multi-cores are closer to enter the evanescent power coupling region. The high sensitivity strain sensors using tapered four-core fibers (FCFs) were demonstrated. The fan-in and fan-out couplers were used to carry out light coupling between singlemode fibers and the individual core of the FCFs. A broadband lightsource from superlumminescent diodes (SLDs) was launched into one of the four cores arranged in a rectangular configuration. When the FCF was substantially tapered, the asymmetric supermodes were produced to generate interferences through this corner-core excitation scheme. During tapering, the supermodes were excited based on a tri-core structure initially and then transited to a rectangular quadruple-core structure gradually to reach the sensitivity of 185.18 pm/μԑ under a tapered diameter of 3 μm. The asymmetric evanescent wave distribution due to the corner-core excitation scheme is helpful to increase the optical path difference (OPD) between supermodes for improving the strain sensitivity.

## 1. Introduction

Over the past few decades, fiber optic sensing technology has been substantially improved to measure strain, pressure, temperature, humidity, vibration, and many other physical parameters with high accuracy or high resolution in harsh environments. Many different kinds of optical fibers have been invented and then played great roles in serving as the sensing devices and information transmission or lasing media. For sensing applications, the polarization, amplitude, phase, and wavelength of the lights in optical fibers can all be modulated or demodulated for delivering or analyzing sensing signals. Among fiber sensors, strain is an important parameter for industrial or scientific applications and is one kind of the essential sensors to monitor the health conditions of the system, e.g., the fatigue situations of the matrix material. Usually, fiber strain can be measured using fiber interferometers or fiber gratings [1,2,3,4,5]. When the sensing areas of the fiber interferometers or gratings are experiencing tensile or compressive stress, the resonant wavelengths will change accordingly. In order to improve the strain sensitivity of the fiber, it is very important to enlarge the optical path difference (OPD) between different modes during elongation of the fiber. To compare with those interferometric strain sensors based on SMFs, the multicore fibers (MCFs) can generally reach better results, since the excited supermodes are assumed by the different high order modes [6,7,8,9,10,11]. In recent years, multicore fibers (MCFs) have been extensively investigated for their important contributions in saving the physical space of the fiber-optic communication systems through enlarging the transmission capacity, which is also called space division multiplexing [12,13,14,15,16,17,18,19,20]. Moreover, active MCFs can also be used to achieve high power fiber lasers based on coherent beam combination [21,22,23,24,25,26]. In addition, it is known that the excited high order modes are very different from that of the traditional singlemode fiber (SMF), since the modes are significantly determined by the dimensions and configurations of all the cores. The supposed mode fields from the individual core generates interferences, which is advantageous to achieve optical sensing with multi-parameters or split optical trapping force [27]. In fusion splicing, in order to match the core positions between the SMF and MCF for delivering optical power with negligible losses, the 7-core MCFs, with one core sitting at the center position, are more popular than other kinds of MCFs to be used to make various kinds of fiber sensors [28,29,30]. However, as a result, the excited supermodes will thus be intrinsically concentric in their fields, and the strain sensors so made are with the strain sensitivity somehow restricted by the limited OPD between supermodes. In contrast, the MCFs, e.g., the FCFs, without central symmetric modes are good options to produce the asymmetric high order supermodes and increase the OPD between them based on the tapered four-core fibers (TFCF) via a corner-core excitation scheme [31]. In FCFs, the four Ge cores can be initially spaced in a situation close to a weakly coupling. By tapering, the four Ge cores were brought more closely spaced so that the evanescent fields of the individual cores overlap each other to turn into a strongly coupling situation. Thus, the asymmetric supermodes can be excited based on the four closely spaced Ge cores when lights are launched into one of the cores.

In this work, we demonstrated high sensitivity strain sensors using TFCF. The experimental set-up is shown in Figure 1. The FCF (Chiral Photonics: SM-4C1500, Pine Brook, NJ, USA) has a cross-sectional microphotograph shown as the inset picture in Figure 1. The core diameter, cladding diameter, distance between the neighboring cores, and distance between the diagonal cores were 8 μm, 50 μm, 74 μm, and 125 μm, respectively. The FCF was spliced with four separated SMFs through the fan-in and fan-out couplers to convey optical power between the SMFs and the individual core of the FCF. The core with the launched lights was defined as the excitation core. Based on this, the core at the diagonal position with respect to the excitation core was defined as the diagonal core. The rest of the two cores sitting beside the excitation core were called neighboring cores. For reading convenience, the excitation core, two neighboring cores, and the diagonal core are designated as c_1_, c_2_, c_3_, and c_4_, respectively, as shown in Figure 1. The weakly coupled FCF was heated by a hydrogen flame and tapered until the supermodes were excited to generate interference. The TFCF was then mounted and fixed on two mechanical clampers of the taper fiber workstation. Then, The SLD lights spanning 1250–1650 nm were launched into c_1_, and the evanescent fields of (c_1_, c_2_, c_3_) and (c_1_, c_2_, c_3_, c_4_) started to overlap each other when D was approaching 30 μm and 25 μm, respectively. Here, D is defined as the diameter of the uniformed tapered region of the TFCF. An optical spectrum analyzer (OSA) (YOKOGAWA AQ6370D, Tokyo, Japan) under an optical resolution of 0.5 nm was used to measure the interference spectra for the TFCF samples. When the TFCF was bilaterally pulled outwardly, the resonant wavelengths blue shifted. The experimental results also reflect that a uniformed tapered diameter D as small as 3 μm can lead to a strain sensitivity of up to 185.18 pm/μԑ. The coefficient of determination R^2^ for those resonant wavelengths versus the strain sensitivity over the strain range 0–730.34 μԑ were all above 0.96 for all the samples. When D was reduced to 3 μm, the R^2^ all reached 0.989, which revealed a very high linearity for the interferometric strain sensors using TFCFs via a corner-core excitation for developing strain sensors with more advanced optical characteristics for practical industrial applications.

## 2. Fabrications, Experimental Set-Up, and Working Principle

As mentioned above, since the MCFs are featured with supermodes interference for improving the sensitivity of sensors, compared with the sensors using conventional SMFs, several papers reported to achieve sensors for temperature, refractive index, and pressure sensing using FCFs [32,33,34,35,36,37]. However, the complicated modes were excited by splicing the FCF and SMF directly. By doing so, the lights launched into the center silica region among the Ge cores, which led to the excitation of chaotic multimodes, not the supermodes supported by the closely-spaced four cores. In contrast, the fiber sensors, especially the strain sensors, using TFCFs with asymmetric field distribution of the supermodes, have not yet been reported. The FCFs used in this work had 4 Ge-doped cores in a rectangular configuration, as shown in Figure 1. The input and output power coupling was realized by splicing the separated SMFs and the individual four cores of the FCF directly through a fan-in and a fan-out coupler, respectively. The SLD lights were launched into one of the 4 cores, named as a corner-core excitation scheme, and which was then defined as the excitation core for determining the excitation orientation. Originally, the launched lights were confined to propagate along the excitation core, shown as core c_1_ in Figure 1. The evanescent fields of the four cores gradually overlapped each other to excite several supermodes for interference when the D was approaching 30 μm during tapering. When D was about 30 μm, the supermodes were generated and observed from the OSA based on the tri-core structure, namely the excitation core c_1_ and its two neighboring cores, c_2_ and c_3_. By continuously decreasing D, the supermodes would be supported and generated based on a rectangular quadruple-core structure, namely the c_1_, c_2_, c_3_, and c_4_. The excited supermode fields were asymmetric due to the corner-core excitation scheme and which is advantageous to make fiber sensors with high sensitivity. This is because the asymmetric modes usually have wider evanescent field distribution compared with the symmetric modes. Moreover, a wider mode field distribution can also correspond to the excitation of higher order modes and a larger OPD along with an increasing propagation length. The OPD can be further enhanced with a reduced D due to the higher order asymmetric supermodes. The detailed working mechanism of the asymmetric supermodes in TFCF is available in our previous work [31]. For strain sensing, the TFCF was then bilaterally fixed by two mechanical clampers and then pulled outwardly to increase the OPD and thus the resonant wavelengths shifted accordingly. In contrast to those strain sensors using hollow core fibers (HCFs), photonic crystal fibers (PCFs), four core fibers (FCFs), multimode fibers (MMFs), inner air cavity, and fiber Bragg gratings (FBGs) [34,38,39,40,41,42], it was experimentally found that, in this work, the strain sensitivity could be obviously improved several tens of times when D was as small as 3 μm. The main reason why the TFCF-based strain sensors using asymmetric supermodes could be superior to those strain sensors using HCFs, PCFs, FCFs, MMFs, FBGs, and so forth, is due to the wider evanescent field distribution for the higher order supermodes. Consequently, the displacement deriving from the tensile strain could efficiently enlarge the OPD between supermodes so as to highly improve the strain sensitivity.

## 3. Results and Discussion

For measurement, the optical resolution (RES) of the OSA was set at 0.5 nm. The spectral responses of the tapered FCF interferometer at different D values are shown in Figure 2a–c. The curves with different colors reflect the input and output power port. For instance, the c_1_–c_1_ and c_1_–c_2_ means the direct-through and cross-coupled states, respectively. It is interesting to note that at D = 26 μm, the spectral responses of the c_1_–c_2_ and c_1_–c_3_ were highly similar and in phase, as shown in Figure 2a. However, the c_1_–c_4_ was clearly out of phase with respect to the c_1_–c_1_ at the wavelengths longer than 1450 nm. This explicitly reveals that the evanescent coupling occurred between c_1_ and c_4_ at the wavelengths longer than 1450 nm. With a much further reduced D, the evanescent coupling between the c_1_ and c_4_ could overlap more tightly, and the interfering spectral oscillations could thus extend to the wavelengths shorter than 1450 nm. Under such circumstance, the asymmetric supermodes would then be entirely supported by the quadruple core configuration. From Figure 2a, the two neighboring cores had the same distance from the excitation core, and thus their spectral responses would be highly similar. On the contrary, the diagonal core was furthest away in terms of the excitation core. The influence of the diagonal core on supermode generation started from the longer wavelengths, and this is the reason why the spectral curve c_1_–c_4_ was out-of-phase to that of the c_1_–c_1_ at the longer wavelength end. This phenomenon disappeared when D was smaller than 20 μm. It is because a reduced distance between cores gave rise to higher coupling coefficients and longer interaction lengths to lead to the over-coupling effect between the asymmetric supermodes. Hence, the synchronous in-phase spectral characteristics for c_2_ and c_3_ were gradually missing with a reduced D. Another reason is that the c_2_ and c_3_ were not stringently identical, so that the coupling coefficients gradually changed with decreasing D as well. From Figure 2b, the average extinction ratios (ER) decreased, whereas the average insertion losses increased for all the curves, compared with that in the Figure 2a. It again can be realized that the four cores were not physically identical in the dimensions and the distance between them. Thus, perfect coupling was becoming more and more difficult to be achieved. However, the main reason was coming from the asymmetric supermodes due to corner-core excitation. On the other hand, with a longer tapered length, the free spectral range (FSR) decreased accordingly, as shown in Figure 2b,c, since a smaller D can give rise to a stronger overlap interaction between four cores for over-coupling situations. Therefore, with the decrease of D of TFCF, the density of the interference fringe increases, and the fringe becomes steeper. Straightforwardly, the OPD will thus be enhanced to improve the strain sensitivity when the TFCF is slightly stretched. From Figure 2c, it is interesting to find that the ER could be significantly improved at some resonant wavelengths. It is believed that the loss of the physical identity between the four cores becomes less effective when D becomes smaller. However, the increased insertion losses are unavoidable due to the excited higher order supermodes at a smaller D. In short, with a reduced D, the FSR and insertion losses are decreased and increased, respectively. The ER may increase contingent upon whether the physical identity between cores is or not maintained. The evolution of the spectral responses in Figure 2a–c reflected the excitation of the supermodes from the tri-core structure at the beginning stage and then transited to the quadruple-core structure as well as the influences of the asymmetry of the supermodes on the spectral characteristics. Obviously, a smaller D can lead to the excitation of higher order supermodes and enhanced OPD for higher sensitivity for strain sensing.

For strain sensitivity measurements, three TFCF samples with D = 15 μm, 7.5 μm, and D = 3 μm were prepared. The corresponding total taper length and uniformed taper length for the three samples were (20, 30, 30.65) mm and (2.8, 2.21, 2.65) mm, respectively. The total taper length was the length containing the uniformed taper length plus the bilateral taper transition length. The bilateral ends of the TFCF were fixed by two mechanical clampers and then stretched by pulling the clampers outward, driven by a stepping motor, with an increment of 5 μm for each step. The spectral responses for D = 15 μm, 7.5 μm, and 3 μm are shown in Figure 3a,c,e in which the resonant dips A, B, C, D blue shifted with increasing tensile stress. The strain can be calculated by the following equation:(1)$$∆\varepsilon \;=\frac{∆L}{L_{0}}\;$$
where $L_{0}$ and ΔL stand for the distance between two clampers, and the displacement of the two clampers [39]. From the Equation (1), the strain can be calculated. The corresponding strains S_A,B,C,D_ for A1, B1, C1, and D1 were 9.99 pm/μԑ, 16.11 pm/μԑ, 10.15 pm/μԑ, 11.53 pm/μԑ when D = 15 μm, and for A2, B2, C2, D2, were 20.5 pm/μԑ, 21.6 pm/μԑ, 21.85 pm/μԑ, and 20.45 pm/μԑ when D = 7.5 μm, and for A3, B3, C3, D3, E3, they were 143.68 pm/μԑ, 108.96 pm/μԑ, 132.5 pm/μԑ, 143.79 pm/μԑ, and 185.18 pm/μԑ when D = 3 μm, respectively, as shown in Figure 3b,d,f. It is interesting to find that the strain sensitivity could be substantially improved when D is reduced to 3 μm. This is because a smaller D can lead to a higher OPD between high order supermodes. However, the spectral characteristics turned out to be more chaotic and more lossy due to the very thin tapered diameter. In addition, the dynamic strain range was reduced to 0–60 μԑ due to the weak mechanical strength. From Figure 3a,c,e, the strain range was 0–730.34 μԑ for D = 15 μm, 0–660 μԑ for D = 7.5 μm, and 0–60 μԑ for D = 3 μm, respectively. Moreover, the cross-coupling states c_1_–c_4_ were observed to achieve the maximum wavelength shift when compared to the other coupling states. It is believed that the OPD can be more efficiently enhanced for the cross-coupling states due to higher order supermodes. For strain sensing, the strain was equal to the displacement (length increment) divided by the distance between two mechanical clampers. Thus, the strain sensitivity is a physical quantity denoted by per unit length. Figure 3b,d,f shows the coefficients of determination R^2^ of linear fitting. The R^2^ is typically higher than 0.989 and can be higher than 0.994 for all the resonant wavelengths over 1500–1650 nm when D is reduced to 3 μm. This strongly reveals that the wavelength shift exhibits a very high linearity, which is very useful for developing the fiber strain sensors with advanced optical characteristics for practical industrial applications. Based on the asymmetric supermodes in a corner-core excitation scheme, the OPD can be substantially improved so that the strain sensitivity is found to be much higher than those interferometers using HCFs, PCFs, FCFs, MMFs, and FBGs, as shown in Table 1 [34,38,39,40,41,42]. The experiments of repeatability of the stain sensing can only be done when a small and precision heating source like the ceramic filament is used in future work.

## 4. Conclusions

In conclusion, we have demonstrated the high sensitivity fiber strain sensors using tapered FCF based on the asymmetric supermode interference via a corner-core excitation scheme. The broadband SLD lights were launched into one of the four cores arranged in a rectangular configuration. With a D of around 30 μm, the supermode interferences were found to be generated based on a tri-core structure, since the spectral characteristics of the diagonal core were partially out-of-phase to the excitation core. Subsequently, with a smaller D, the supermode interferences occurred based on a rectangular quadruple-core core structure. The resonant dip wavelengths blue shifted with increasing tensile stress, and the best strain sensitivity was up to 185.18 pm/μԑ under a tapered diameter of 3 μm. This tapered FCF interferometer via a corner-core excitation was firstly demonstrated and is highly promising to make high sensitivity strain sensors ascribing to the asymmetric supermodes.

## Figures and Tables

**Figure 1 micromachines-13-00431-f001:**
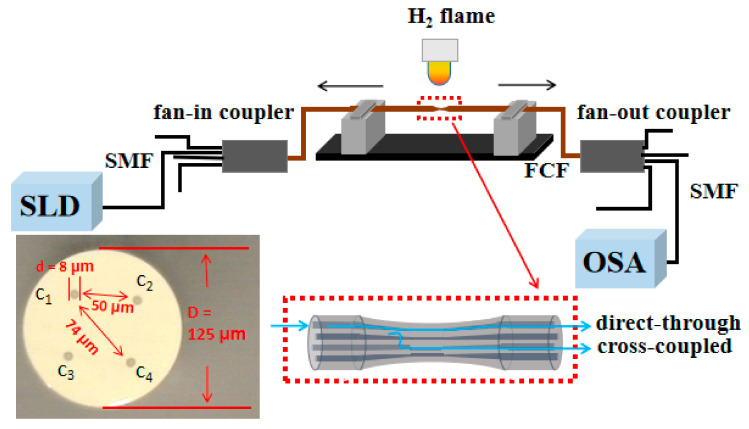
Experimental set-up of the tapered FCF interferometer in which c_1_, c_2_, c_3_, c_4_ represent the excitation core, two neighboring cores, and diagonal cores, respectively. The inset picture shows the designation of the four cores of the un-tapered FCF under 1000 × CCD microscope.

**Figure 2 micromachines-13-00431-f002:**
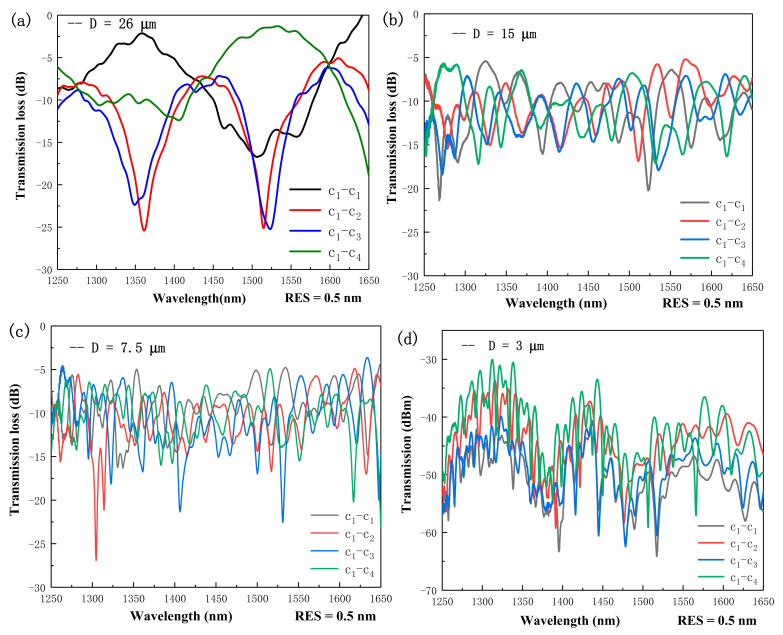
Spectral responses of the oscillating curves for (**a**) D = 26 μm, (**b**) D = 15 μm, (**c**) D = 7.5 μm, and (**d**) D = 3 μm, for the c_1_–c_1/2/3/4_ coupling states.

**Figure 3 micromachines-13-00431-f003:**
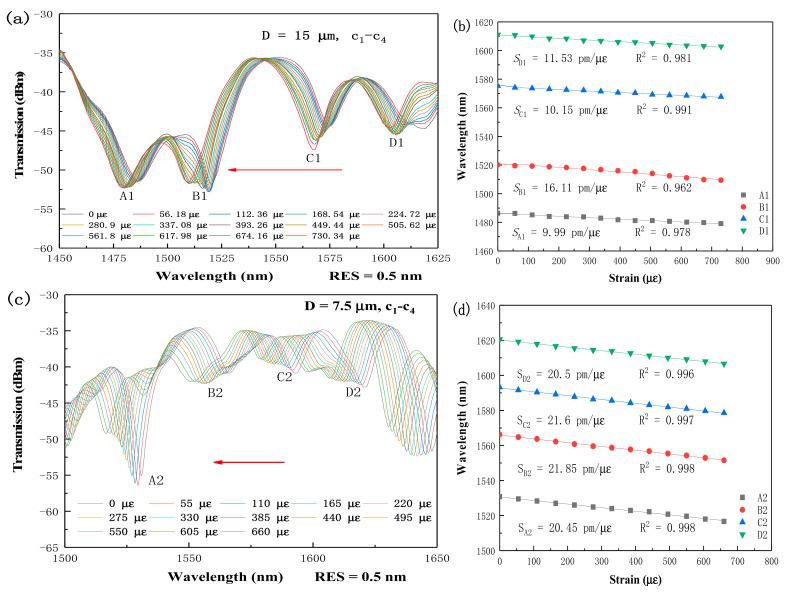
Spectral responses of the TFCF interferometer under different tensile strain for (**a**) D = 15 μm, (**c**) D = 7.5 μm, and (**e**) D = 3 μm, respectively. Linear fitting curves of the dip wavelength shifts and the coefficients of determination R^2^ of linear fitting for (**b**) D = 15 μm, (**d**) D = 7.5 μm, and (**f**) D = 3 μm, respectively.

**Table 1 micromachines-13-00431-t001:** Summary of strain sensors using different fiber.

Reference No.	This Work	[34]	[38]	[39]	[40]	[41]	[42]
Sensor scheme	TFCFs	FCFs	HCFs	PCFs	MMFs	Inner air-cavity	FBGs
Dynamic range	0–60 με	0–2000 με	0–1000 με	0–2100 με	0–1000 με	0–2000 με	0–300 με
Maximum sensitivity	185.18 pm/μԑ	1.78 pm/με	6.80 pm/με	2.10 pm/με	2.6 pm/με	22.5 pm/με	39.791 pm/με

## Data Availability

The data that support the findings of this study are available upon request from the corresponding author. The data are not publicly available due to privacy or ethical restrictions.

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
