# Peer review of "High Sensitivity Strain Sensors Using Four-Core Fibers through a Corner-Core Excitation"

_micromachines, 2022, doi:10.3390/mi13030431_

Round 1

Reviewer 1 Report

Please see attached

Author Response

Reviewer1

The author proposed the high sensitivity strain sensors using the tapered four-core fibers (FCFs). There are still some unclear or wrong points, please correct them according to the suggestions as follows:

Point 1: Does the taper length of the four-core fibers impact on the measurement sensitivity? What are the considerations for choosing the optimal taper length in spectral analysis, it seems to need clarification, and what is the taper length in this experiment (in case of D=7.5 μm).

Response: The total taper length for the samples (D = 7.5 μm) is 30 mm, in which the uniformed tapered length is 2.21 mm. The total taper length is the length containing the uniformed taper length plus the bilateral taper transition length. For strain sensing, the strain was measured by calculating the length increment over the distance between two mechanical clampers. So, the strain sensitivity is the physical quantity calculated by per length unit. Thus, the taper length is not very crucial to the strain sensitivity, especially when the fiber between clampers is remained uniformed and homogeneous. We have added the explanations at Line 193. They read as “The corresponding total taper length and uniformed taper length for the three samples are (20, 30, 30.65) mm and (2.8, 2.21, 2.65) mm, respectively. The total taper length is the length containing the uniformed taper length plus the bilateral taper transition length.” Also at Line 217 and they read as “For strain sensing, the strain was equal to the displacement (length increment) divided by the distance between two mechanical clampers. So, the strain sensitivity is a physical quantity denoted by per unit length.”

Point 2: When D=7.5 μm, it seems that these four core fibers will gather together. How is the repeatability of the spectral measurement of the components for different fabrications? Please explain it.

Response: This is a good question. In fact, it is very difficult to make identical tapered fiber samples at different batches of tapering, even for the very mature fiber couplers manufacturing process. The experiments can be done when a small and precision, but very expensive, heating source like ceramic filaments is used. So, we are sorry that we could not provide the experimental results about repeatability. We have added the sentence “The experiments of repeatability of the stain sensing can only be done when the small and precision heating source like the ceramic filament is used in the future work.” at Line 227.

Point 3: As shown in the inset picture of Figure 1, please provide the schematic diagram of D=7.5 μm, the original single core fiber (d=8 μm) should also be changed on the original diameter under the condition of D=7.5 μm.

Response: We are sorry to make the reviewer misunderstood. In fact, the inset picture of Figure 1 is the cross-sectional microphotograph under 1000x CCD microscope. The cross-sectional microphotograph of the sample (D= 7.5 um) is too small to be handled. In order to avoid misunderstanding, we have revised the figure caption as “The inset picture shows the designation of the four cores of the un-tapered FCF under 1000x CCD microscope.”

Point 4: There are some errors in the description of Figure 3, please correct them as follows: --- different tensile strain for (a) D = 15 μm and (c) D = 7.5 μm respectively. ----linear fitting for (b) D = 15 μm and (d) D = 7.5 μm, respectively.

Response: Thanks for the notifications! We have corrected all of them as requested.

Point 5: In Line213-216, the authors mentioned that the achieved results are better than previous research results. It is recommended to present the current measurement results and the related publish technologies in a comparison table, including sensing element technology, dynamic

Response: We thank the reviewer for the useful suggestion. We have added Table 1 in line 236 and in which we have also cited a few more references for comparison.

Reviewer 2 Report

Please clearly define D as in line line 88 it was introduced without prior clarification.

The sensitivity is 21.85 21pm/μÔ‘ for a tapered diameter of 7.5 μm. It this the optimum dimension as the authors have experimented with 26, 15 and 7.5 μm. Since 7.5 μm is the smallest attempted will making it smaller achieve better sensitivity or is there an optimal dimension?

The caption for Fig 3 is incorrect "Fig. 3. Spectral responses of the TFCF interferometer under different tensile strain for (a) D = 15 μm and (b) D = 7 μm, respectively." It is should be (a) and (c) instead of (a) and (b). Further (b) and (d) are the linear fit but the caption mentioned  (c) and (d).  Please correct these.

How does this work stacked up with the work by similar researcher?  What is the benefit and improvement in performances?

Author Response

Reviewer2

Please clearly define D as in line line 88 it was introduced without prior clarification.

Response: We added the definition of D at line 84. They read as " Here, D is defined as the diameter of the uniformed tapered region of the TFCF."

The sensitivity is 21.85 21pm/μÔ‘ for a tapered diameter of 7.5 μm. It this the optimum dimension as the authors have experimented with 26, 15 and 7.5 μm. Since 7.5 μm is the smallest attempted will making it smaller achieve better sensitivity or is there an optimal dimension?

Response: We thank the reviewer for this interesting question. We did not try the tapered diameter below 7.5 μm since the fiber becomes so brittle and fragile under such a small diameter. However, as requested, we have tried to make another sample with tapered diameter of 3 μm. Surprisingly, the strain sensitivity can be significantly increased to 185.18 pm/μÔ‘. However, the dynamic strain range is reduced to 0 - 60 μÔ‘, as a trade off. We have added the new results. Regarding the optimized tapered diameter, we can only say D = 3 μm is the best results in our work. Probably, a smaller tapered diameter could give rise to a higher strain sensitivity but we can not achieve the goal based on our limited resources. We have added the explanations and they read as “and for A3, B3, C3, D3, E3, were 143.68 pm/μÔ‘, 108.96 pm/μÔ‘, 132.5 pm/μÔ‘, 143.79 pm/μÔ‘, and 185.18 pm/μÔ‘ when D = 3 μm, respectively, shown in Figures 3b,d,f. It is interesting to find that the strain sensitivity can be substantially improved when D is reduced to 3 μm. This is because a smaller D can lead to a higher OPD between the high order supermodes. However, the spectral characteristics turn out to be more chaotic and more lossy due to the very thin tapered diameter. In addition, the dynamic strain range is reduced to 0 - 60 μÔ‘ due to the weak mechanical strength.” At Line 206.

The caption for Fig 3 is incorrect "Fig. 3. Spectral responses of the TFCF interferometer under different tensile strain for (a) D = 15 μm and (b) D = 7 μm, respectively." It is should be (a) and (c) instead of (a) and (b). Further (b) and (d) are the linear fit but the caption mentioned (c) and (d).  Please correct these.

Response: Thanks for pointing out the errors. We have corrected them as requested.

How does this work stacked up with the work by similar researcher?  What is the benefit and improvement in performances?

Response: We have added the table 1 for comparison. In this work, the strain sensitivity can be significantly improved to 185.18 pm/μÔ‘ when the tapered diameter is done to 3 μm, which is several tens of times higher than those works in references 40-44. Obviously, the asymmetric supermodes in the tapered four-core fibers can be helpful to substantially improve the OPD between supermodes as well as the strain sensitivity. We have added the sentences “Based on the asymmetric supermodes in a corner-core excitation scheme, the OPD can be substantially improved so that the strain sensitivity is found to be much higher than those interferometers using HCFs, PCFs, FCFs, MMFs, and FBGs, shown in table 1 [34,40-44].” at Line 224.

Reviewer 3 Report

In this paper, the authors demonstrated a high-sensitivity strain sensor by using tapered four-core fibers (TFCF). Overall, this is quite an interesting paper and provides a new strategy to fabricate high-sensitivity strain sensors. Some minor revisions can be made to further strengthen the manuscript.

  1. The mechanism of using TFCF to achieve higher sensitivity than the existing methods needs to be discussed.
  2. What is the range of strain that this sensor can detect?
  3. Is there any possibility to make a demo device by utilizing this sensitive sensor?

Author Response

Reviewer3

In this paper, the authors demonstrated a high-sensitivity strain sensor by using tapered four-core fibers (TFCF). Overall, this is quite an interesting paper and provides a new strategy to fabricate high-sensitivity strain sensors. Some minor revisions can be made to further strengthen the manuscript.

  1. The mechanism of using TFCF to achieve higher sensitivity than the existing methods needs to be discussed.

Response: The working mechanism of the asymmetric modes in TFCF can be available in our previous publication and we have added the explanation “The detailed working mechanism of the asymmetric supermodes in TFCF can be available in our previous work [39].” at Line 129. In comparison with the existing methods, we have added the explanations “The main reason why the TFCF-based strain sensors using asymmetric supermodes can be superior to those strain sensors using HCFs, PCFs, FCFs, MMFs, FBGs, and so forth, is due to the wider evanescent field distribution for the higher order supermodes. Conse-quently, the displacement deriving from the tensile strain can efficiently enlarge the OPD between supermodes so as to highly improve the strain sensitivity.At Line 136.

  1. What is the range of strain that this sensor can detect?

Response: In our work, we had tried the dynamic range of strain over 0 - 730.34 μÔ‘ contingent upon the tapered diameter. A smaller tapered diameter can lead to a weaker mechanical strength and thus the dynamic strain range is restricted. From Figures 3a,c,e, the strain range is 0 - 730.34 μÔ‘ for D = 15 um, 0 - 660 μÔ‘ for D = 7.5 um, 0 - 60 μÔ‘ for D = 3 um. We have added the sentence “From Figures 3a,c,e, the strain range is 0 - 730.34 μÔ‘ for D = 15 μm, 0 - 660 μÔ‘ for D = 7.5 μm, 0 - 60 μÔ‘ for D = 3 μm, respectively.” At Line 213.

  1. Is there any possibility to make a demo device by utilizing this sensitive sensor?

Response: This is a very good suggestion. In fact, our tapered fibers were mounted on the mechanical clampers on the fiber taper work station. The clampers can be precisely moved by the stepping motors with a step resolution of 1 μm. So, our sensors are not far to the demo device in practical applications, except for the package. With a good design of package, the strain sensors can be demonstrated for commercialization. We will consider to do it in the near future. We thank the reviewer for this comment!

Round 2

Reviewer 1 Report

The author has made complete corrections to my concerns and suggestions, and overall, the current version is acceptable for publication

Reviewer 2 Report

The authors have provided additional experimental results and this is very helpful.  Coupled with reasonable explanation of the new results, in my opinion, the manuscript should be accepted for publication. 

Perhaps a final proof read by a native English speaker will help to polish the manuscript.